# Thyroid Nodule Characterization: Which Thyroid Imaging Reporting and Data System (TIRADS) Is More Accurate? A Comparison Between Radiologists with Different Experiences and Artificial Intelligence Software

**DOI:** 10.3390/diagnostics15162108

**Published:** 2025-08-21

**Authors:** Emanuele David, Lorenzo Aliotta, Fabrizio Frezza, Marianna Riccio, Alessandro Cannavale, Patrizia Pacini, Chiara Di Bella, Vincenzo Dolcetti, Elena Seri, Luca Giuliani, Mattia Di Segni, Gianmarco Lo Conte, Giacomo Bonito, Antonino Guerrisi, Fabio Mangini, Francesco Maria Drudi, Corrado De Vito, Vito Cantisani

**Affiliations:** 1Department of Medical Surgical Sciences and Advanced Technologies “GF Ingrassia”, University Hospital Policlinic “G. Rodolico-San Marco”, 95123 Catania, Italy; lore.aliotta@gmail.com; 2Department of Information Engineering, Electronics and Telecommunications, “Sapienza” University of Rome, 00184 Rome, Italy; fabrizio.frezza@uniroma1.it; 3Department of Public Health and Infectious Diseases, Sapienza University of Rome, 00185 Rome, Italy; marianna.riccio@uniroma1.it (M.R.); corrado.devito@uniroma1.it (C.D.V.); 4Department of Radiology, Anatomical-Patology and Oncology, Sapienza University of Rome, 00162 Rome, Italy; alessandro.cannavale@hotmail.com (A.C.); patry.shepsut91@gmail.com (P.P.); chiara.dibella@uniroma1.it (C.D.B.); vincenzodolcetti@gmail.com (V.D.); eliseri2499@gmail.com (E.S.); mattia.di.segni@gmail.com (M.D.S.); gianmarco.loconte@uniroma1.it (G.L.C.); giacomo.bonito@alice.it (G.B.); francescom.drudi@uniroma1.it (F.M.D.); vito.cantisani@uniroma1.it (V.C.); 5Department of Biotechnological and Applied Clinical Sciences, San Salvatore Hospital, University of L’Aquila, Vetoio Stree, 67100 L’Aquila, Italy; luca.giuliani@uniroma1.it; 6Radiology and Diagnostic Imaging Unit, Department of Clinical and Dermatological Research, San Gallicano Dermatological Institute IRCCS, Via Elio Chianesi 53, 00144 Rome, Italy; antonino.guerrisi@ifo.it; 7Department of Engineering, Niccolò Cusano University, 00166 Rome, Italy; fabio.mangini@uniroma1.it

**Keywords:** thyroid cancer characterization, CAD system, TIRADS

## Abstract

**Purpose:** This study aimed to compare: the performance of K-TIRADS, EU-TIRADS and ACR TIRADS when used by observers with different levels of experience compared with the gold standard of cytology, and to evaluate the diagnostic performance of CAD (computer-aided design) compared with TI-RADS systems. **Methods and Materials:** In total, 323 thyroid nodules were evaluated in patients who were candidates for needle aspiration. Three observers with different levels of experience evaluated the diagnostic accuracy of three risk stratification systems (ACR TI-RADS, EU-TIRADS and K-TIRADS) and CAD software (S-Detect, made by Samsung) in characterizing the nodules. The results were compared with cytology examination. All nodules were characterized in terms of shape, margins, composition, calcifications, size, echogenicity and microcalcifications, and by stratifying individual nodules by using the three TIRADS systems; then S-detect software was applied and the data were compared with each other and with the gold standard. **Results:** Through cytology, 308 benign and 33 malignant nodules were identified. ACR-TIRADS showed a sensitivity of 100%, a specificity of 86%, a positive predictive value of 43% and a negative predictive value of 100%. EU-TIRADS showed a sensitivity of 100%, a specificity of 79%, a positive predictive value of 33% and a negative predictive value of 100%. K-TIRADS showed a sensitivity of 100%, a specificity of 89%, a positive predictive value of 50% and a negative predictive value of 100%. S-Detect combined with EU-TIRADS showed a high agreement (>95%) with the gold standard. **Conclusions:** K-TIRADS’s positive predictive power was slightly better than the other TIRADS, suggesting greater accuracy in correctly diagnosing positive cases. S-DETECT combined with EU-TIRADS has similar results to S-Detect with ACR- and K-TIRADS in terms of sensitivity, specificity and negative predictive power. However, it has a slightly better positive predictive power, suggesting greater accuracy in correctly diagnosing positive cases than the ACR- and K-TIRADS classification systems. In general, S-Detect cannot yet be considered a substitute for the human observer but only as an important support for human evaluation and an excellent and fast help to provide a comprehensive and complete report. **Clinical Relevance/Application:** S-Detect is a valuable tool for characterizing thyroid nodules when integrated with radiologist evaluation. It is also an important support tool for less experienced observers. Particularly interesting is the approach of use in integrated combination of the K-TIRADS by the human observer with S-Detect using EU-TIRADS, which could increase the overall diagnostic efficiency of the systems.

## 1. Introduction

A thyroid nodule is defined as a focal lesion, distinct from the surrounding healthy thyroid parenchyma, and can be recognized by imaging or histological sampling [1,2].

Most thyroid nodules are benign; however, between 10 and 15% of these are malignant [3,4]. According to the 2020 report of the World Cancer Observatory, thyroid cancer is responsible for approximately 586,000 cases worldwide [5].

For this reason, all appreciable thyroid nodules should be individually and carefully examined to assess the presence or absence of ultrasound features capable of predicting the likelihood of benign or malignant noduled and thus obtain indications of the best diagnostic-therapeutic attitude: follow-up necessary, follow-up negligible, or, if necessary, minimally invasive diagnostics in depth.

In recent decades, thanks to improved technology and modern knowledge in diagnostic imaging, there has been an increase in the ability to detect thyroid nodules and consequently thyroid cancers [6].

Ultrasound is the primary imaging modality used in the study of nodular thyroid disease and allows clinicians to recognize and evaluate some suggestive features of malignancy; on the basis of these, the need for further diagnostic investigations should be assessed, such as fine needle aspiration (FNA) assessment.

Currently, the vanguard of literature is represented by TIRADS systems proposed by three different scientific societies, including the Korean TIRADS (K-TIRADS) published in 2011 by the Korean Society of Thyroid Radiology (KSThR) and revised in 2016; the ACR-TIRADS, where ACR stands for American College of Radiology, which published its score in 2016; and the EU-TIRADS, where EU stands for European, proposed in 2017 [7,8,9].

Recently, radiomics is taking its place in thyroid imaging and beyond; it uses extraction algorithms to derive various quantitative features from radiological images, and can be used by machine learning (ML) systems, which is a subset of artificial intelligence (AI) from which deep learning (DL) is derived. These innovative technologies may eventually translate into software used directly by clinicians, namely computer-aided diagnosis (CAD) [10,11] (Figure 1a–c), which is a type of deep learning software.

The aim of this study was to compare, through a retrospective analysis, the performance of the various TIRADS ultrasound systems mentioned above (K-TIRADS, EU-TIRADS and ACR-TIRADS) when each one is used by observers with different levels of experience compared with the actual malignancy rate obtained from a standard cytological/histological examination, including the time required for their application. Finally, we also performed a statistical analysis comparing the diagnostic performance of AI (CAD) with that of a human observer, looking for the possible presence of diagnostic discrepancy depending on the degree of experience of the human observer, to understand how S-Detect software can really help the less experienced radiologist.

## 2. Methods and Materials

In our retrospective study, which was approved by the local Ethics Committee (Comitato Etico territoriale Lazio Area 1), with approval number 7458, referring to protocol 1011/2023 and meeting minutes 20 December 2023, we included 277 patients, for a total of 334 thyroid nodules selected for fine needle aspiration (FNA), who had previously signed informed consent.

The 277 patients included came to our institute for observation in the period between September 2020 and October 2023, and all selected 334 thyroid nodules were submitted to cytological evaluation by FNA. Nodules that were found to be benign by the FNA cytological evaluation were rechecked at 18 and 36 months to confirm their stability and to consider them as such [12]. In contrast, the nodules that were found to be malignant or undetermined at cytology were subjected to surgical resection and subsequent histological examination in accordance with the Italian classification of thyroid cytology. Patients with more than three thyroid nodules, cysts and nodules smaller than 5 mm were excluded from our study.

All images related to the included nodules were obtained at diagnosis or follow-up control or during cytologic (FNA) sampling by an experienced radiologist with more than 25 years of experience in thyroid ultrasound, using a high-frequency (14–20 GHz) linear probe. Images were acquired in B-Mode and stored for subsequent evaluation according to the standardization criteria suggested by the TIRADS system. Regarding the quality of the images obtained, standardization was made possible by the processing work of the S-Detect software, which automatically forced us to discard the images that did not reflect the quality criteria.

Cytologic analysis of the nodules included in the FNA assessment was obtained within 15–20 days of sample collection. Each specimen was fixed in formalin, specially stored and sent to the Pathologic Anatomy Department of our institute for case analysis. For the evaluation of thyroid cytology, we used the new classification (TIR) published by the Italian societies of endocrinology (AIT, AME and SIE) and that of of pathological anatomy and cytology (SIAPEC-IAP) in 2014 [13]. The data obtained were stored in a database for later comparison with the ultrasound findings.

Three observers with different experience levels (low level: <5 years of experience; medium level: between 5 and 15 years of experience; high level: >15 years of experience) were recruited for our retrospective analysis. The three different observers were not aware of the patients’ clinical information, except for age, the databases obtained from other observers, the echographic data extrapolated from CAD, nor the cytological results of FNA.

Each of the three observers viewed images of 334 thyroid stored nodes in B-Mode and, for each, applied the three risk stratification systems (ACR-TIRADS, EU-TIRADS and K-TIRADS) by assigning a score for each of five ultrasound characteristics (composition, echogenicity, shape, margins, calcification or targeted echogenic foci) and then obtaining a TIRADS risk category from the sum of these. Later, the same images were processed by CAD software (S-Detect). All data obtained were compared with each other and with the gold standard.

All data and imaging results assigned by each observer and the CAD were stored separately in order to allow subsequent comparison through appropriate statistical analysis with the results of the cytological examination.

Radio/cytological agreement was estimated by comparing the TIRADS score assigned for each node by each TIRADS system (EU-TIRADS, K-TIRADS, ACR-TIRADS) and the cytological TIR score obtained from the corresponding FNA sample. On the basis of the TIRADS systems, nodules classified as 2 or 3 were considered in our study to be benign, and those classified as 4 or 5 as malignant. Regarding TIR classification, nodes with cytology scores of TIR2 were considered benign, while the others (TIR3, -4 and -5) were considered malignant; specifically TIR3 nodes were considered indeterminate for malignancy (TIR3a: low-risk malignancy; TIR3b: high-risk malignancy). The category TIR1 corresponded to those not valid for FNA sampling. Radio/cytological data, obtained for each thyroid node included, were compared with the gold standard to estimate the sensitivity, specificity, positive predictive value (PPV), negative predictive value (NPV) and area under the curve (AUC), each with 95% confidence intervals (CI).

The sample size was calculated assuming a Type I error (α) of 5%, an expected prevalence of 10% and a marginal error of 5% to ensure an expected sensitivity of the expert observer of at least 98%. On the basis of these parameters, the required sample size was estimated to be 334 individuals.

The results obtained by individual observers and S-Detect, and their comparison with the final cytohistological diagnosis, have been compiled in Table 1, Table 2 and Table 3.

## 3. Statistical Analysis

An in-depth statistical analysis was carried out to assess inter-observer concordance at different levels.

The primary objective of the study was to evaluate inter-observer concordance for each of the three different TIRADS system included (EU-, K- and ACR-TIRADS) when the TIRADS score was assigned by observers with different levels of experience (low, <5 years; medium, between 5 and 15 years; and high, >15 years). Furthermore, inter-observer agreement was investigated for each of the 5 sonographic items considered (composition, form, calcifications, echogenicity and margins). Then, the results obtained by each of the three human observers were compared with those calculated using the CAD software (S-Detect).

Finally, radio/cytological agreement was evaluated for each TIRADS system (K, EU and ACR) when applied by observers with different levels of experience, comparing the TIRADS scores assigned with the TIR (the Italian cytology classification for thyroid nodes) cytological results from FNA.

To confirm the intra-observer agreement, one of the three recruited observers reviewed the same nodules a second time from the first view of each nodule; in this regard, to homogenize the sample, the observer with a medium level of experience was chosen (between 5 and 15 years).

Inter-observer agreement for TIRADS scores was evaluated both among all readers (more than two observers) and between each observer and the S-Detect (two observers) system for ACR-, K- and EU-TIRADS using the multi-user weighted Cohen’s kappa.

In addition, by means of the multi-user Cohen’s kappa, it was also possible to obtain data on the frequency of presentation of each of the five ultrasound items studied (composition, shape, margins, calcifications and echogenicity).

We considered statistically significant p-values to be below 0.05. All statistical analyses were carried out with the help of SPSS version 21 statistical software (SPSS Inc., Chicago, IL, USA).

In this regard, it was decided to establish the degree of inter-observer agreement on the basis of the interpretation of Landis and Koch, thus identifying the following degrees of concordance: poor agreement for Cohen’s k values of <0.2, fair agreement for k values of 0.2–0.4, moderate agreement for k values of 0.4–0.6, substantial agreement for k values of 0.6–0.8 and excellent for k values > 0.8 [14].

In contrast, the inter-observer concordance was not evaluated regarding the nodules’ dimensions because the set of images had been obtained and archived previously and therefore did not have an inter-observer variability parameter of interest.

## 4. Results

The sample of patients from which we selected the 334 nodules included in our study, was composed mainly of female subjects, precisely 234 women against 43 males, for a total of 277 patients. The mean age was 49.2 years (SD = 16.4), with median values ranging from 45.3 years (SD = 10.3) in the female cohort to 59.5 years (SD = 13.8) in the male cohort.

Cytological data on 334 thyroid nodules sampled showed that 258 nodules were benign or non-malign (specifically, all were in Category TIR2,no TIR1 nodules observed), 33 nodules were suspected of malignity (all in Category TIR4; no TIR5), while the remaining 43 nodules were cytologically indeterminate (TIR3, all of which were TIR3a, considered to be at a low risk of malignity) as shown in Table 3. The average diameter of the nodules examined was about 26.9 mm (DS = 1.01).

### 4.1. Agreement Among Human Observers with Different Levels of Experience

Results regarding the inter-observer concordance in assigning a certain TIRADS score at each nodule by the three human observers recruited showed different Cohen’s k values depending on the TIRADS system considered (K, ACR and EU). Among all three human readers, inter-observer agreement was substantial (k = 0.624) for ACR and moderate both for EU (k = 0.542) and for K (k = 0.496), as shown in Table 4.

As regard the score assigned to each of the five sonographic characteristics, inter-observer agreement among observers with different levels of experience (high, average and low) showed extremely variable Cohen’s k values depending on the different sonographic parameters considered (Table 5).

For the parameter composition, agreement was excellent for ACR and EU (k = 0.826 and k = 0.809, respectively) and substantial for K (k = 0.785).

For shape, agreement resulted to be substantial for ACR (k = 0.793), EU (k = 0.716) and K (k = 0.687).

Regarding echogenicity, concordance was moderate for ACR (k = 0.498), EU (k = 0.441) and K (k = 0.389).

For calcifications or targeted echogenic foci, agreement was from moderate to fair with values of k = 0.416 for ACR, k = 0.318 for EU and k = 0.351 for K.

For the last parameter considered, margins, poor agreement was registered for the three TIRADS systems included, with values of k = 0.134 for ACR, k = 0.119 for EU and k = 0.106 for K.

### 4.2. Agreement Between Human Observers with Different Levels of Experience and S-Detect

Regarding the observer/S-Detect concordance, we also recorded different Cohen values, which varied depending on the level of experience of the observer (high, medium or low) considered in the comparison with S-Detect and the TIRADS system applied (whether K, ACR or EU).

When considered the ACR-TIRADS system, the observer/S-Detect agreement was from substantial to moderate with values of k = 0.762 for the high-level observer, k = 0.654 for the medium-level observer and k = 0.596 for the low-level observer.

A similar degree of agreement was found for K-TIRADS (from substantial to moderate) with k values of k = 0.679 for the high-level observer, k = 0.603 for the medium-level observer and k = 0.536 for the low-level observer.

For EU-TIRADS, instead, observer/S-Detect agreement was from moderate to fair; in particular, Cohen values registered were k = 0.417, k = 0.334 and k = 0.295 for the observers with high, medium and low levels of experience, respectively. All the Cohen’s k values above are summarized in Table 4.

### 4.3. Radio/Cytological Agreement

As regards the radio/cytological concordance, intended to show the correspondence between the TIRADS ultrasound category calculated (benign or malign) and the assigned TIR cytological class (benign, malign or specifically indeterminant for malignancy), it was evaluated by different statistical indices and showed very variable results, depending on various factors such as, for example, the type of observer (whether human or S-Detect), the different levels of experience of the human observers (high, medium or low) or the type of TIRADS system (EU, K, ACR) applied, as we summarise in Table 2.

When we considered the human observer with a high level of experience, radio/cytological concordance recorded sensitivity (SEN) values of 100% for EU, ACR and K; also, the negative predictive value (NPV) was 100% for all three TIRADS systems. Positive predictive value (PPV) was 50% for ACR, 50% for K and 36.7% for EU. With respect to specificity (SPE), the values were 89% for ACR-TIRADS, 85.7% for K-TIRADS and 75.8% for EU-TIRADS, while the AUC data recorded values of 94.5% for ACR, 87.9% for EU and 92.8% for K.

The observer with an average experience level recorded SEN values of 63.6 for ACR-, K- and EU-TIRADS. NPV was 95.2% for ACR-, 95.1% for K- and 94.6% for EU-TIRADS. PPV showed values between 24.7% for K- and ACR-TIRADS and 23.1% for EU. On the other hand, for specificity and AUC, they had values of 78.8% and 71.1 for ACR-TIRADS; 76.7% and 70.2% for K-TIRADS; and 70.4% and 67% for EU-TIRADS, respectively.

For the observer with a low experience level, radio/cytological concordance registered SEN values of 60.6% for ACR, K and EU, and 94.7% (ACR), 94.6% (K) and 94.4% (EU) for NPV. PPV values varied between 9.9% of EU and 22.2% of ACR. SPE had values of 76.7% for ACR-TIRADS, 75.4% for K-TIRADS and 54.4% for EU-TIRADS; while the values of AUC were of 68.7% for ACR, 57.5% for EU and 68% for K.

The concordance as regard the results obtained with S-Detect showed the same value of SEN for K-TIRADS, ACR-TIRADS and EU-TIRADS, which is 66.7%. NPV was 96.2% for both EU and K and 96.3% for ACR TIRADS, while PPV was 50% (EU and K) and 66.7% (ACR). S-Detect had a SPE of 92.7% for EU and K and 96.3% for ACR. Finally, the AUC of S-Detect was 79.7% both for EU and K, and 81.5% for ACR.

## 5. Discussion

Statistical analyses carried out on data obtained from 334 nodules included from 277 patients with thyroid disease confirmed the expected results presented to date in the literature [15,16], showing a better degree of agreement, from substantial to excellent, regarding sonographic items such as nodule composition (k = 0.826) and shape (k = 0.783), whereas there was a moderate to poor degree of concordance with the other three echographic characteristics such as echogenicity (k = 0.498), margins (k = 0.134), and presence or absence of calcifications or targeted echogenic foci (k = 0.396). So, the parameter margins of the nodules resulted to be the main factor of inter-observer discordance, especially in nodules with a negative cytological result (TIR3).

Regarding the concordance between human observers and S-Detect when the three different TIRADS systems were applied, our study has substantially confirmed the data in the literature [17], showing that observer/S-Detect agreement was better for ACR, slightly higher than EU, with optimal median Cohen’s k values of k = 0.762 and k = 0.679, respectively, while it was poor for K-TIRADS (k = 0.417) for the high-level observer. In our data, we also observed homogeneously decreasing concordance values for all three TIRADS systems when the comparison was made between S-Detect and human observers with decreasing level of experience (from high to low).

We also undertook a more in-depth analysis by calculating the degree of observer/S-Detect concordance for each of the five ultrasound parameters used in the three TIRADS systems in order to exclude discrepancies that are more dependent on the ultrasound characteristic considered than on the TIRADS system used. Our results showed excellent to substantial agreement for nodule composition and shape (k = 0.8–0.5), moderate to fair for the margins (k = 0.5–0.4) and fair to poor regarding the echogenicity and calcifications (k = 0.1–0.3).

In a comparison of the TIR cytological class result from FNA for each nodule and the different TIRADS scores assigned depending on the type of observer or TIRADS system considered, data on radio/cytological concordance point out that when TIRADS systems were applied by human observers, the best match was with ACR-TIRADS, as it had optimal diagnostic accuracy and slightly higher than other systems such as K and EU.

Radio/cytological concordance for each of the three TIRADS systems was shown to vary in a way that was closely dependent on the different levels of experience of the three radiologists: the observer with more than 15 years’ experience (high level) showed better accuracy than the less experienced observers with between 5 and 15 years (medium level) or less than 5 years (low level) of experience, who also had lower and almost overlapping accuracy values.

In particular, the observer with a high level of experience recorded values of sensitivity (SEN) of 100% for all three types of TIRADS and, therefore, there were no subjects with a “false negative”. To confirm the certainty of the negative result, the negative predictive value (NPV) was also 100%, so all patients assumed to be healthy were found to be healthy. The positive predictive value (PPV) was 50% concerning the low prevalence of malignancy, meaning that the patients included had a 50% chance of being ill if the ultrasound examination was positive. The relative specificity (SPE) values are 89% for ACR-TIRADS, 85.7% for K-TIRADS and 75.8% for EU-TIRADS; this implies a certain proportion of “false positives”, especially with the use of EU TIRADS (lower SPE). This is associated with a not perfect but still very high diagnostic accuracy with values of 94.52% for ACR, 94.52% for EU and 92.86% for K.

Observers with medium and low experience levels reported lower SEN values of, respectively, 64.9% and 60.1%, with the values almost comparable for the use of the three TIRADS systems, indicating the presence of a fair proportion of “false negative” patients. Even the NPV, although slightly lower, remains optimal, with values of 91.3% and 88.9%, respectively, indicating that most patients considered healthy were indeed so. There was no disagreement compared with the observer with a high level of experience regarding PPV (that is, the probability that the patients included would be ill), for which the value remained at 50%. The SPE confirmed the better diagnostic certainty of ACR-TIRADS (SPE = 78.8% for the average-level observer and SPE = 76.7% for the low-level observer) compared with K (SPE = 76.7% for the average-level observer and SPE = 75.4% for the low-level observer) and EU (SPE = 70.4% for the average-level observer and SPE = 54.4% for the low-level observer). Diagnostic accuracy was low compared with the observer with a high level of experience or with S-Detect and had values of 84.69% for ACR, 83.64% for EU and 80.95% for K.

Radio/histological concordance among the three TIRADS systems (K, ACR and EU) when applied by S-Detect software did not show substantial differences in the degree of concordance, contrary to the human observers.

Diagnostic accuracy assessed for the results obtained with S-Detect showed, as already mentioned, an almost perfect concordance among the K-TIRADS, ACR-TIRADS and EU-TIRADS systems.

SEN for S-Detect’s measurement is 66.7% for all three TIRADS systems; this results in a moderate number of “false negative” subjects. NPV is 96.2% for both ACR and K and 96.3% for EU TIRADS, so the patients’ probability of being healthy is high. PPV is 50% (ACR-K) and 66.7% (EU). SPE is higher than SEN: it is 92.7% (EU and K) and 96.3% (ACR). This is associated with fewer “false positives” and with a greater diagnostic certainty than human observers’ measurements, regardless of the different levels of observer experience. Diagnostic accuracy was 79.68% for EU, 81.51% for ACR and 79.68% for K: slightly lower compared with the human observer with a higher level of experience but better than the observers with medium and low levels of experience.

It emerges, therefore, that S-Detect software can help human observers with medium/low levels of experience in improving specificity for the TIRADS grade awarded; this is also supported in a recent study by Lee S.E. et al., which demonstrated that S-Detect improves the diagnostic accuracy of the youngest or most inexperienced radiologists by using a model based on the reader’s self-learning process [18].

In conclusion, our study confirmed once more a better diagnostic accuracy of ACR TIRADS system than K and EU-TIRADS when applied by each of the three human observers with different experience level; while no significant discrepancy was observed when applied by S-Detect software. It was also found that ACR-TIRADS was better in the observer/S-Detect agreement (k = 0.7624).

The degree of radio/cytological concordance calculated for each of the three human observers with different experience level sand the S-Detect software was evaluated at different levels and shows that regarding the diagnostic accuracy, the human observer was better than S-Detect when she/he had a high level of experience, while she/he was worse when we considered the dataset obtained from the other two observers with medium and low levels of experience. So, we can presume that S-Detect is an innovative software tool, easy to use in the evaluation of nodular thyroid disease, providing excellent reliability and a good degree of concordance in comparison with the human observer, representing a valid aid for the radiologist, especially if he/she is young and inexperienced; in addition, S-Detect has greater diagnostic specificity compared with the human observer.

## Figures and Tables

**Figure 1 diagnostics-15-02108-f001:**
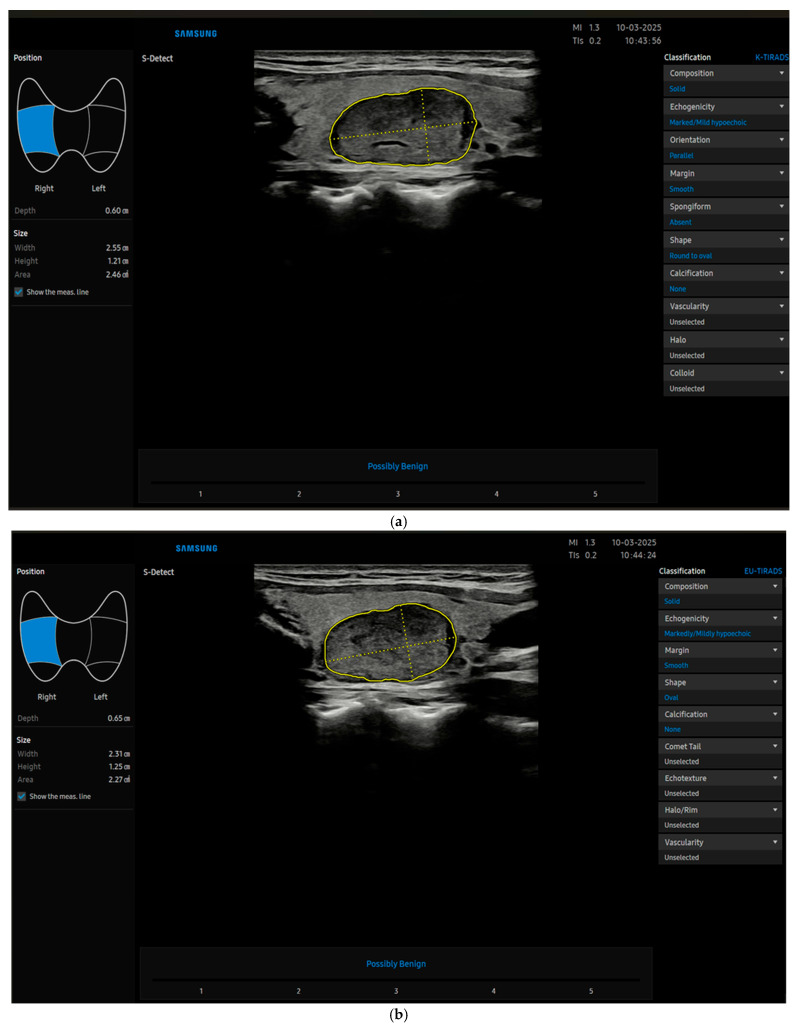
(**a**) The image shows a thyroid nodule characterized by CAD (S-Detect) as benign (later confirmed cytologically and by follow-up) by using K-TIRADS. (**b**) The image shows the same thyroid nodule characterized by CAD (S-Detect) as benign by using EU-TIRADS (later confirmed cytologically and by follow-up). (**c**) The image shows the same thyroid nodule characterized by CAD (S-Detect) as benign by using ACR-TIRADS (later confirmed cytologically and by follow-up).

**Table 1 diagnostics-15-02108-t001:** Intervariability in TIRADS scores among observers with different levels of experience (high, average, low) or S-Detect when applying different TIRADS systems (ACR, EU and K).

**ACR**	**High-Level Observer**		**Average-Level Observer**		**Low-Level Observer**		**S-Detect Observer**	
	Frequency	Percentage	Frequency	Percentage	Frequency	Percentage	Frequency	Percentage
**TIRADS 1**	0	0	0	0	0	0	66	19.76
**TIRADS 2**	116	34.73%	105	31.43%	100	29.94%	148	44.31
**TIRADS 3**	142	42.51%	126	37.72%	128	38.32%	76	22.75
**TIRADS 4**	43	12.87%	60	17.96%	62	18.56%	33	9.88
**TIRADS 5**	33	9.89%	43	12.87%	44	13.17%	11	3.29
**EU**	**High-Level Observer**		**Average-Level Observer**		**Low-Level Observer**		**S-Detect Observer**	
	Frequency	Percentage	Frequency	Percentage	Frequency	Percentage	Frequency	Percentage
**TIRADS 1**	0	0	0	0	0	0	22	6.58
**TIRADS 2**	64	19.16%	45	13.47%	24	7.18%	128	38.32
**TIRADS 3**	172	51.49%	183	54.79%	196	58.68%	151	45.20
**TIRADS 4**	54	16.16%	57	17.06%	61	18.26%	11	3.29
**TIRADS 5**	44	13.17%	49	14.67%	53	15.86%	22	6.58
**K**	**High-Level Observer**		**Average-Level Observer**		**Low-Level Observer**		**S-Detect Observer**	
	Frequency	Percentage	Frequency	Percentage	Frequency	Percentage	Frequency	Percentage
**TIRADS 1**	0	0	0	0	0	0	22	6.58
**TIRADS 2**	53	15.86%	57	17.06%	66	19.76%	127	38.02
**TIRADS 3**	215	64.37%	194	58.08%	179	53.59%	141	42.21
**TIRADS 4**	22	6.58%	32	9.58%	34	10.17%	33	9.88
**TIRADS 5**	44	13.17%	51	15.26%	55	16.46%	11	3.29

**Table 2 diagnostics-15-02108-t002:** Radio/cytological agreement for human observers with different levels of experience (high, average and low) and S-Detect considering different TIRADS systems (ACR, EU and K).

**High-Level Observer** **Cytology/Radiology**	**ACR**		**EU**		**K**	
	**POS.**	**NEG.**	**POS.**	**NEG.**	**POS.**	**NEG.**
**Normal**	33	268	57	179	33	197
**Abnormal**	33	0	33	0	33	0
**TOTAL**	66	268	90	179	66	197
**SEN (95% CI)**	100% (89.4–100)		100% (89.4–100)		100% (89.4–100)	
**PPV (95% CI)**	50% (37.4–62.6)		36.7% (26.8–47.5)		50% (37.4–62.6)	
**NPV (95% CI)**	100% (98.6–100)		100% (98–100)		100% (98.1–100)	
**SPE (95% CI)**	89% (84.9–92.3)		75,8% (69.9–81.2)		85.7% (80.4–89.9)	
**ACC (95% CI)**	94.5% (92.8–96.3)		87.9% (85.2–90.7)		92.8% (90.6–95.1)	
**Average-Level Observer** **Cytology/Radiology**	**ACR**		**EU**		**K**	
	**POS.**	**NEG.**	**POS.**	**NEG.**	**POS.**	**NEG.**
**Normal**	64	237	89	212	70	231
**Abnormal**	21	12	21	12	21	12
**TOTAL**	85	249	110	224	91	243
**SEN (95% CI)**	63.6% (45.1–79.6)		63.6% (45.1–79.6)		63.6% (45.1–79.6)	
**PPV (95% CI)**	24.7% (16–35.3)		19.1% (12.2–27.7)		23.1% (14.9–33.1)	
**NPV (95% CI)**	95.2% (91.7–97.5)		94.6% (90.8–97.2)		95.1% (91.5–97.4)	
**SPE (95% CI)**	78.7% (73.7–83.2)		70.4% (64.9–75.5)		76.7% (71.6–81.4)	
**ACC (95% CI)**	71.2% (62.5–79.8)		67% (58.3–75.8)		70.2% (61.5–78.9)	
**Low-Level Observer** **Cytology/Radiology**	**ACR**		**EU**		**K**	
	**POS.**	**NEG.**	**POS.**	**NEG.**	**POS.**	**NEG.**
**Normal**	70	231	183	218	74	227
**Abnormal**	20	13	20	13	20	13
**TOTAL**	90	244	203	231	94	240
**SEN (95% CI)**	60.6% (42.1–77.1)		60.6% (42.1–77–1)		60.6% (42.1–77.1)	
**PPV (95% CI)**	22.2% (14.1–32.2)		9.9% (6.1–14.8)		21.3% (13.5–30.9)	
**NPV (95% CI)**	94.7% (91.1–97.1)		94.4% (90.6–97)		94.6% (90.9–97.1)	
**SPE (95% CI)**	76.7% (71.6–81.4)		54.4% (49.3–59.3)		75.4% (70.1–80.2)	
**ACC (95% CI)**	68.7% (59.9–77.5)		57.5% (48.7–66.3)		68% (59.2–76.8)	
**S-Detect Observer** **Cytology/Radiology**	**ACR**		**EU**		**K**	
	**POS.**	**NEG.**	**POS.**	**NEG.**	**POS.**	**NEG.**
**Normal**	11	290	22	279	22	279
**Abnormal**	22	11	22	11	22	11
**TOTAL**	33	301	44	290	44	290
**SEN (95% CI)**	66.7% (48.2–82)		66.7% (48.2–82)		66.7% (48.2–82)	
**PPV (95% CI)**	66.7% (48.2–82)		50% (34.6–65.4)		50% (34.6–65.4)	
**NPV (95% CI)**	96.3% (93.6–98.2)		96.2% (93.3–98.1)		96.2% (93.3–98.1)	
**SPE (95% CI)**	96.3% (93.6–98.2)		92.7% (89.1–95.4)		92.7% (89.1–95.4)	
**ACC (95% CI)**	81.5% (73.3–89.7)		79.7% (71.4–88)		79.7% (71.4–88)	

**Normal**, determinate at cytology (benign (TIR1/2) or malignant (TIR 4/5)); **abnormal**, indeterminate at cytology (TIR3a/b); **POS**., positive at radiology (indeterminate (TIRADS 3) or malignant (TIRADS 4/5)); **NEG**.; negative at radiology (benign, TIRADS 1/2); **SEN**, sensibility; **PPV**, positive predictive value; **NPV**, negative predictive value; **SPE**, specificity; **ACC**, diagnostic accuracy.

**Table 3 diagnostics-15-02108-t003:** Cytological results obtained by FNA.

Cytology (FNA)	Frequency	Percentage
**TIR1**	0	0
**TIR2**	258	77.24%
**TIR3a**	43	12.87%
**TIR3b**	0	0
**TIR4**	33	9.88%
**TIR5**	0	0
**TOTAL**	334	100%

**Table 4 diagnostics-15-02108-t004:** Inter-observer agreement among human observers and S-Detect in assigning different TIRADS scores.

Inter-Observer Agreement	ACR	EU	K
**Among all three human observers**	k = 0.624	k = 0.542	k = 0.496
**Between the high-level observer and S-Detect**	k = 0.762	k = 0.417	k = 0.679
**Between the average-level observer and S-Detect**	k = 0.654	k = 0.334	k = 0.603
**Between the low-level observer and S-Detect**	k = 0.596	k = 0.295	k = 0.536

k, multi-user weighted Cohen’s kappa.

**Table 5 diagnostics-15-02108-t005:** Inter-observer agreement among human observers for each sonographic characteristic.

Human Observers’ Agreement	ACR	EU	K
**Composition**	k = 0.826	k = 0.809	k = 0.785
**Shape**	k = 0.793	k = 0.716	k = 0.687
**Echogenicity**	k = 0.498	k = 0.441	k = 0.389
**Margins**	k = 0.134	k = 0.119	k = 0.106
**Calcifications Or Targeted Echogenic Foci**	k = 0.416	k = 0.318	k = 0.351

k, multi-user weighted Cohen’s kappa.

## Data Availability

The raw data supporting the conclusions of this article will be made available by the authors on request.

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
