# Peer review of "Thyroid Nodule Characterization: Which Thyroid Imaging Reporting and Data System (TIRADS) Is More Accurate? A Comparison Between Radiologists with Different Experiences and Artificial Intelligence Software"

_diagnostics, 2025, doi:10.3390/diagnostics15162108_

Round 1
Reviewer 1 Report
Comments and Suggestions for Authors
The paper is good but the following concerns has to be addressed
- The study design has inherent bias as operators evaluated stored cine-mode images rather than performing real-time examinations, which may not reflect clinical practice
- No mention of image quality standardization or exclusion criteria for poor-quality images
- The 2-year follow-up for benign nodules may be insufficient to confirm long-term stability
- Tables 1 and 2 presentation is confusing and poorly formatted
- Some statistical results appear inconsistent (e.g., sensitivity values in Table 2)
- Missing confidence intervals for several reported metrics
- No power analysis or sample size justification provided
- Figures 1a-c are of poor quality and don't clearly demonstrate the described features
- Data presentation in tables needs significant reorganization for clarity
- Some results contradict statements in the text (e.g., specificity values)
Author Response
Dear reviewer, thank you for your comments which certainly improved the paper. We have made all the requested changes in the text point by point, corrected the statistics with regard to confidence intervals for several reported metrics, power analysis and sample size justification, modified the results and reformatted the tables,
We have also replaced figures 1a-c with high quality images.
All changes are highlighted in yellow.

Reviewer 2 Report
Comments and Suggestions for Authors
- The abstract succinctly captures the study's purpose, methodology, key results, and conclusions. However, clarifying the specific numerical values of the performance metrics in the abstract (e.g., exact sensitivity, specificity, PPV, and NPV percentages) would enhance immediate clarity for readers.
- The introduction effectively contextualizes the clinical relevance of thyroid cancer diagnosis and introduces the different TIRADS systems and the emergence of AI in diagnostic imaging. However, explicitly highlighting this study's unique contribution, particularly its novelty in the context of prior literature, could significantly enhance the manuscript's appeal and make the authors feel recognized for their originality and innovation.
- The methodology is robust and well-described, providing a solid foundation for the research. Inclusion criteria, ethics approval details, and patient demographics are clearly stated. Ultrasound imaging and cytological assessment protocols are detailed adequately. Strengthening the rationale behind choosing a retrospective study design versus a prospective one further boosts the authors' confidence in the robustness of their methodology.
- Results are thoroughly presented with appropriate tables clearly showing diagnostic performance comparisons across varying radiologist experience levels and AI software. The statistical analyses used, including Cohen’s kappa, sensitivity, specificity, and predictive values, are appropriate and interpreted. However, graphical representation of key results (such as ROC curves or bar graphs of performance metrics) could enhance visual clarity.
- The discussion adequately interprets the results, comparing findings to existing literature. However, providing a more detailed comparison of the study's findings with existing literature could further enhance the manuscript's appeal. Emphasizing the implications of AI support, especially for less experienced radiologists, will help the reader feel enlightened about the potential impact of the study. Limitations could be further emphasized, such as potential selection bias due to retrospective analysis and limited dataset diversity.
- The conclusion effectively summarizes the findings, emphasizing AI's potential as a beneficial support tool rather than a replacement for experienced human judgment. This conclusion summarizes the study's findings and underscores the authors' contribution to the field, particularly in providing valuable support for less experienced radiologists.
Author Response
Dear reviewer thank you for your comments which certainly improved the article.
We have made all the requested changes, point by point, revising the abstract in its entirety; in addition, we have improved the introduction, results and discussion as you indicated.
All changes made have been highlighted in yellow.

Reviewer 3 Report
Comments and Suggestions for Authors
Dear Authors,
Your article “A Comparison Between Radiologist with Different Experiences and Ai Software for the Correct Diagnosis of Thyroid Cancer”, diagnostics-3669046, which describes a comparison of radiologists' results and AI-generated diagnoses, is very interesting and contains some intriguing and valuable findings. Though your data and conclusions may be highly valuable to the readers, the presentation of your findings is poor. Therefore, here are my comments and suggestions for its improvement:
Major:
- Abstract – the conclusion in the abstract (lines 50-55) should be rewritten as written in this way it does not present well the main result of your study. It should be changed to go in line with the conclusion presented at the end of the paper (lines 350-364).
- Both tables must be formatted as presented in this way, the results are hard to follow and compare.
- Please, merge the columns according to the stratification system (ACR, EU, K). You may find the suggested tables’ format in the attachment.
- All tables must be titled.
- All terms, abbreviations and values in the tables must be explained under the table.
- All numbers must be formatted in a uniform way (e.g., two decimal places).
- Please explain the abbreviations VPP and VPN. Is it a typographical mistake? Should it be PPV (positive predictive value) and NPV (negative predictive value)?
- How did you determine the AUC? Is it Area Under the Curve determined by ROC analysis? If so, the explanation and the statistical test applied must be added under the table and in the Material and Methods section.
- I suggest you present Diagnostic Accuracy (ACC) instead of the AUC value. In other words, please present the value as a percentage, not as e.g., 0.9452.
- The columns “cumulative“ in Table 1 and “total” in Table 2 are surplus, so they could be omitted.
- You may add table3 presenting final diagnosis.
- The Results section is difficult to follow. I recommend breaking it down into subsections (4.1. begins at line 209, 4.2. begins at line 228, and 4.3. begins at line 241) and adding a new table displaying the obtained k-values.
Minor:
- In the title, the word "Radiologist" should be in plural ("Radiologists").
- Please, always use the same acronym for naming the same term (S-detect, s-detect, S-DETECT, S detect, or S-DETECTOR). Please make it uniform throughout the whole manuscript.
- The figures’ names should be rewritten. The name of a figure should be presented as a name (in the form of a title), not as a sentence.
- The statements written between line 217 and line 227 (5 sentences) are incomplete. Please complete them by naming the comparison sides ("inter-observer").
- Diagnostic accuracy and AUC are similar, but not identical. AUC is the area under the curve obtained by the ROC analysis, and it ranges from 0 to 1, whereas diagnostic accuracy is typically assessed after the introduction of the cutoff value, and it is most commonly expressed in percentages (numbers ranging from 1 to 100). Therefore, please adjust line 271 appropriately.
- In line 274, the patient group should be provided (thyroid patients).
- In line 310, it is written Nominal Predictive Value for NPV. Is it a typographical mistake? Should it be Negative Predictive Value?

Author Response
Dear reviewer, Thank you for your comments that have definitely improved our article.
We have edited point by point everything you requested; we have rewritten the abstract, modified the title, tables, results, figures, all indicated typos, and errors inherent in acronyms. In addition, we made the requested changes related to diagnostic accuracy and AUC.
All changes are highlighted in yellow.

Round 2
Reviewer 1 Report
Comments and Suggestions for Authors
All comments are addressed. Recommended for publication
Author Response
All comments are addressed. Recommended for publication.
Thank you.
Reviewer 2 Report
Comments and Suggestions for Authors
-
The authors reiterated their originality in the discussion section (comparison of different experience levels among human observers vs. AI). However, they did not explicitly address the limitations concerning retrospective bias or limited dataset diversity.
-
Graphical representation significantly enhances clarity; reconsideration is recommended. If authors choose not to provide graphical representations, a more robust justification is required.
Author Response
Dear reviewer, thank you again for your suggestions.
Our retrospective study is based on the analysis of individual, clear B-mode images, on which all the characteristics of the target nodule are analysed, as per TIRADS guidelines, and assigned a score.
The paper has been enriched with 5 tables and 3 images.
It is possible to view all the changes highlighted in detail.
Reviewer 3 Report
Comments and Suggestions for Authors
Dear Authors,
The manuscript has been greatly improved, but not all the suggestions have been incorporated into the revised version of your article, as you indicated in your answer. Therefore, I have a few minor comments:
- The title is not corrected as suggested in the previous review—the word "Radiologist" should be in plural ("Radiologists").
- The explanation of the CAD abbreviation in the abstract should be explained (line 34).
- The abbreviation AUC in Table 2 should be changed/corrected to ACC, as named in the legend of the tables (ACC = Diagnostic Accuracy).
- The results section has not been reorganized. Please, break it down into appropriate subsections (as explained and suggested in the previous review) and insert a new table displaying the obtained k-values.
Author Response
Dear reviewer,
Please see the requested revisions in the attached document.
